# Implementation and first report of the Brazilian Kidney Biopsy Registry

Irene L. Noronha[1,2]*, Rodrigo José Ramalho[1,3], Claudia Maria Costa de Oliveira[1,4], Marilia Bahiense-Oliveira[1], Fabricio Augusto Marques Barbosa[1], Jose de Resende Barros Neto[1], Ronny Mitsuoka[5], Osvaldo Merege Vieira-Neto[6], Precil Diego Miranda de Menezes Neves[1,2,7], on behalf of the Brazilian Kidney Biopsy Registry Working Group[1¶]

1 Clinical Nephrology Department of the Brazilian Society of Nephrology, São Paulo, Brazil, 2 Renal Division, Hospital das Clínicas, University of São Paulo Medical School, São Paulo, Brazil, 3 Division of Nephrology, Sao Jose do Rio Preto Medical School, Sao Paulo, Brazil, 4 Renal Division, Hospital Universitário Walter Cantídio, Fortaleza, Brazil, 5 Postgraduate, Radiology Institute Hospital das Clinicas, University of São Paulo Medical School, São Paulo, Brazil, 6 Nephrology Discipline, Ribeirão Preto Medical School—University of São Paulo, Ribeirão Preto, Brazil, 7 Nephrology and Dialysis Center, Hospital Alemão Oswaldo Cruz, São Paulo, Brazil

¶ Membership of The Brazilian Kidney Biopsy Registry Working Group is provided in the Acknowledgments.
* irenenor@usp.br

## Abstract

### Background

Kidney biopsy registries are valuable tools for guiding clinical practice and developing health policies. In 2021, the Brazilian Society of Nephrology (SBN) created the Brazilian Kidney Biopsy Registry (BKBR). This is the first BKBR report, presenting patient data from 2021.

### Methods

BKBR is a web-based platform hosted on the BSN website, which contains patient demographics, clinical data, frequency, and distribution of histologic diagnosis of Brazilian adult native kidney biopsies.

### Results

Of the 1012 cases registered in 2021, 954 cases were evaluated after excluding pediatric and kidney transplant cases. Twenty-one centers enrolled patients, with representation from all Brazilian regions. There was a slight predominance of females (52.6%), a mean age of 44.7 ± 16 years, and 13.6% of patients were >65 years old. The main indication for kidney biopsy was renal dysfunction (56%) and nephrotic syndrome (41.4%), respectively. At the time of the biopsy, 47.9% of the patients were hypertensive and 15.2% were diabetic. Although 66.2% of patients had eGFR ≤60ml/min/1.73m² upon biopsy, the majority (60.2%) had mild interstitial fibrosis and tubular atrophy. The most frequent diagnosis in the BKBR was glomerular disease (74.8%). Lupus nephritis was the most frequent diagnosis of glomerular disease (22.6%), followed by IgA nephropathy (13%) and focal segmental glomerulosclerosis (12.2%).

**Data Availability Statement:** All relevant data are reported in the paper and its Supporting information file.

**Funding:** The author(s) received no specific funding for this work.

**Competing interests:** ILN has received honoraria for Steering Committee roles, scientific presentations and/or advisory board attendance from Travere, Chinook, Vertex, Roche, and AstraZeneca. In addition, The George Institute for Global Health holds research contracts for trials in kidney disease. OMVN has received speaker's honoraria from GSK, Takeda and AstraZeneca. All the other authors declare no conflict of interest.

## Conclusion

This is the first report of a Nationwide registry of kidney biopsies in Brazil. This data provides pivotal information about the kidney disease profile in this country with continental dimensions.

## 1. Introduction

Glomerular diseases are relevant causes of chronic kidney diseases (CKD) worldwide, representing the second or third cause of renal replacement therapy in many countries (ANZDATA 45th report 2022; 2020 USRDS Annual Data Report ERA Registry Annual Report 2021). Data from the last 5 annual Brazilian Dialysis Surveys show that glomerular diseases are the third cause of renal failure in the dialysis population in Brazil (9% of all causes) [1, 2]. However, until recently there was no national Registry of Glomerular Diseases in Brazil.

Studies on the epidemiology of glomerular diseases in Brazil are scarce and mostly restricted to reports from specific nephrology centers. One of the largest studies in this area was the Paulista Registry of Glomerulonephritis, published in 2006, with a 5-year period of data from centers in São Paulo State, the most populous state in the country [3]. In this report, focal segmental glomerulosclerosis (FSGS) (29.7%), membranous nephropathy (MN) (20.7%), and IgA nephropathy (IgAN) (17.8%) ranked as the most frequent primary glomerular diseases. Similar results were reported in a study from São Paulo Federal University in 2010 [4] describing FSGS (24.6%), IgAN (20.1%), and MN (20.7%) as the most prevalent primary glomerulonephritis A report in 2017 from a center in Pernambuco State, in the northeast region of Brazil, described FSGS (43%), MN (15%), Minimal Change Disease (MCD) (14%), and IgAN (9%) as the most frequent glomerular diseases in the study [5].

The establishment of the nationwide Brazilian Kidney Biopsy Registry (BKBR) as a prospective database with greater precision and accuracy will provide better understanding of the epidemiology of glomerular diseases in Brazil. Herein we present the results of the first year of this registry.

## 2. Methods

This is a multicenter, prospective, observational national registry, conducted by the Clinical Nephrology Department of the Brazilian Society of Nephrology (BSN). Nephrology and Nephropathology centers from all Brazilian regions were invited to register in the Brazilian Kidney Biopsy Registry (BKBR). A web-based registry system was developed, and data of biopsied patients were registered by each center. In 2021, the first year of the operation of this web platform, 21 centers, representing all Brazilian regions, participated in this registry by prospectively inputting the data relative to native kidney biopsies.

The BKBR study was approved by the Ethics Committee of the Faculty of Medicine of São José do Rio Preto. CAAE: 59947022.6.0000.5415. Informed consent was waived because of the retrospective nature of the study and the analysis used anonymous clinical data. The data were extracted from web-database of BKBR by a data-scientist from Brazilian Society of Nephrology on 01-Sep-2022.

### 2.1. Development of the platform and the questionnaire

A web platform for the registry was constructed with technical characteristics to guarantee safety and efficiency, enabling to record cases with relevant demographic, clinical and

pathological information, in addition to the histologic diagnosis of kidney biopsies, and open to modifications, incorporation of other data sources and future protocols.

The web-based data management system was developed to support the database (MYSQL) with encryption-based internet security protocol (SSL). To create the BRKB website, we purchased web hosting with a Linux server from Locaweb. We then registered the domain www.glomerulopatias-sbn.org.br on registro.br, pointing it at Locaweb's DNS, and installed the SSL certificate for security. We developed and tested the site using Visual Studio Code and the following languages: HTML; PHP; JavaScript; CSS and SQL. We saved it on the web host and then created some users and passwords for accessing the form and filling in and sending the data.

The questionnaire was elaborated by the nephrologist members of the Clinical Nephrology Department of the BSN. A relatively simple questionnaire was designed, aiming to collect the main relevant clinical, laboratory and histologic information (**S1 Table**). Information of biopsied patients may be registered on the website by nephrologists or renal pathologists of each center. No identifiable information of patient identity was used on the web platform. The date of birth, the initial letters of the name and sex of the patients were the information recorded for identification. A list of the diagnoses is provided to ensure ease of input by the user, and completing most fields required only one click. The web-platform generates an Excel spreadsheet including all necessary parameters for analyses. The data added to the system is annually evaluated and validated. Incorrect or discrepant data is checked with each center that registered the patients for confirmation or correction.

BRKB data is public and belongs to the Brazilian Society of Nephrology. In order to have access to BKBR data, the researcher must fill in an online form with the required information. This request will be evaluated by the BSN scientific board, which will make the data available to the researcher. All research and publications derived from BKBR data must be approved by an ethics committee and the BSN must be cited as a partner institution.

## 2.2. Patient population and data source

Data available in the registry from patients aged 18 years or over, who underwent kidney biopsy of native kidneys from January 1st to December 31st, 2021 were included in this cohort. Biopsies from pediatric and transplant patients were excluded. Kidney biopsies were performed as clinically indicated by the patients' nephrologists. In cases with more than one kidney biopsy, only the one which provided the diagnosis was considered for the analysis. In cases of sequential biopsies, only the first one was included.

The following patient demographic data reported were analyzed: age, gender, race (in Brazil this is subjectively cited), and Brazilian region of the patient. Information concerning clinical manifestations that are usually an indication of renal biopsy such as nephrotic syndrome, non-nephrotic proteinuria, hematuria, rapid progressive glomerulonephritis (RPGN), and kidney dysfunction were requested.

Serum creatinine at the time of the kidney biopsy was required and CKD-EPI equation without race was automatically calculated by the platform [6]. In addition, information on hypertension, diabetes, systemic and family renal diseases were also requested.

## 2.3. Histological diagnoses

The histological diagnosis of kidney biopsies was based on light microscopy, immunofluorescence, and electron microscopy whenever available. The number of glomeruli and semiquantitative estimation of interstitial fibrosis and tubular atrophy (IFTA) were registered as severe

(> 50%), moderate (25–50%) or mild (<25%). Histologic diagnosis could be selected from a list presented in the registry (**S1 Table**).

## 2.4. Categorization of kidney diseases

Kidney biopsies of this cohort were grouped into six main categories (**S2A Table**): glomerular diseases, diabetic nephropathy, tubulointerstitial diseases (acute interstitial nephritis, acute tubular necrosis (ATN), and IgG4-related kidney disease), monoclonal gammopathies, vascular diseases (herein including hypertensive nephrosclerosis and systemic sclerosis) and hereditary inherited kidney diseases, as well as a miscellaneous group (including sickle cell disease, uric acid nephropathy, chronic tubulointerstitial nephritis), and an unclassifiable group of biopsies.

Additionally, in order to more specifically analyze kidney diseases with glomerular involvement in this cohort, all cases with predominant kidney involvement were regrouped and analyzed (**S2B Table**) In this additional category, along with all diseases included in the glomerular diseases category, diabetic nephrotic, amyloidosis, monoclonal immunoglobulin deposition disease (MIDD), proliferative glomerulonephritis with monoclonal IgG deposits (PGNMID), immunotactoid and fibrillary glomerulopathies (grouped as Monoclonal Gammopathies), and collagen type IV disorders, LCAT Deficiency and Fabry's Disease (Inherited Kidney Diseases) were specifically analyzed.

## 2.5. Statistical analysis

Statistical analyses were performed using the SPSS statistical software package for Windows (Version 29.0, IBM Corp). Continuous data were presented as mean and standard deviation and compared using the Student's t-test or ANOVA. Non-parametric data were expressed as median and interquartile ranges and compared using Mann-Whitney U-test or Kruskal-Wallis test. Categorical values were expressed as absolute frequencies and percentages. Differences were tested using the chi-squared test or Fisher's exact test. The base layer of the maps were retrieved from https://www.ibge.gov.br/geociencias/downloads-geociencias.html. A $p$-value < 0.05 was considered statistically significant.

## 3. Results

### 3.1. Patient population and regional representation

A total of 1012 kidney biopsies performed in 2021 were included in the registry in the first year of the BKBR. After excluding rebiopsies cases and patients with age < 18 years, a total of 954 biopsies were included in the analysis. Twenty-one centers participated, with representation from all Brazilian regions. Regarding the frequency of patients by Brazilian regions, there was a predominance of the Southeast region (63.5%), followed by the Northeast (24.7%), South (8.8%), Midwest (2.7%) and North (0.3%). The distribution of the number of registered patients by state and the percentage of registered patients by region are shown in **Fig 1**.

### 3.2. Patient demographics

The median age of all patients at kidney biopsy was 44.7 ± 16.0 years (ranging from 18 to 93 years), with a peak incidence in the 21–50 year (51%). It is noteworthy that 13.4% of kidney biopsies were from patients with age ≥65 years and 1.5% with age >80 years. **Fig 2** illustrates the distribution of patients according to age.

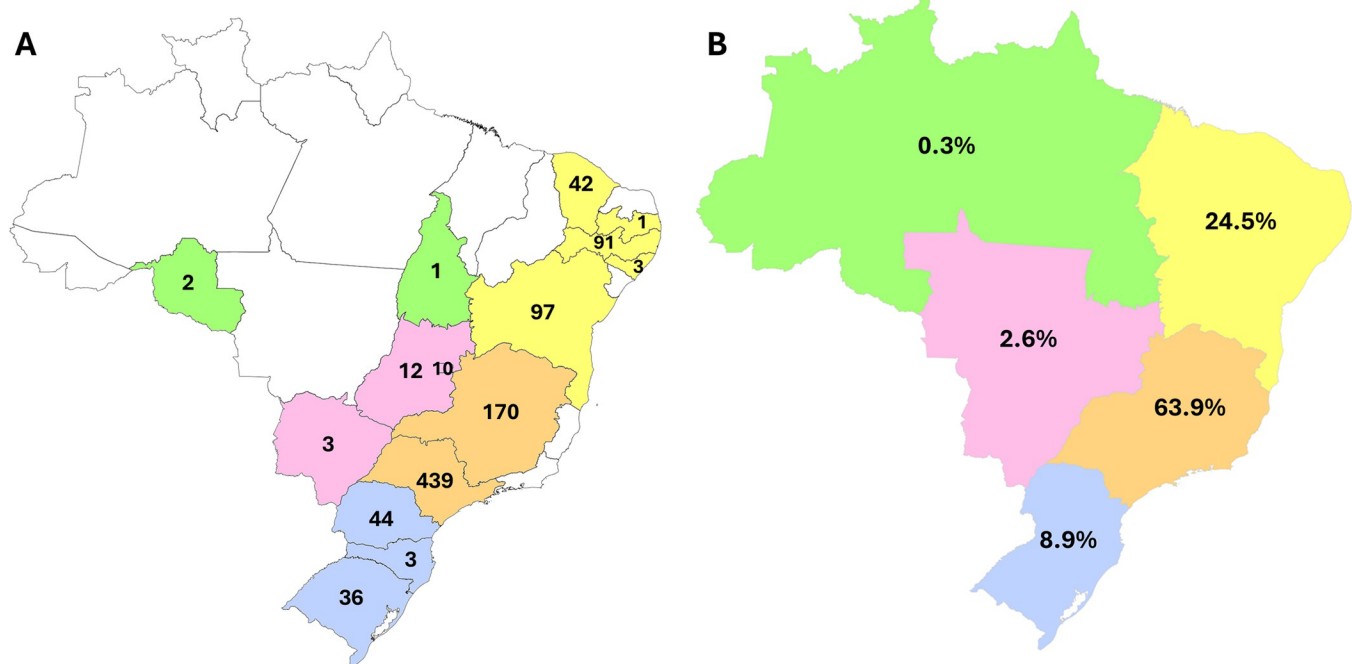

**Fig 1.** Distribution of the number of renal biopsies registered in the BKBR by geographic region of Brazil (A) per State, (B) percentage in the different geographical regions. The base layer of the maps were retrieved from https://www.ibge.gov.br/geociencias/downloads-geociencias.html.

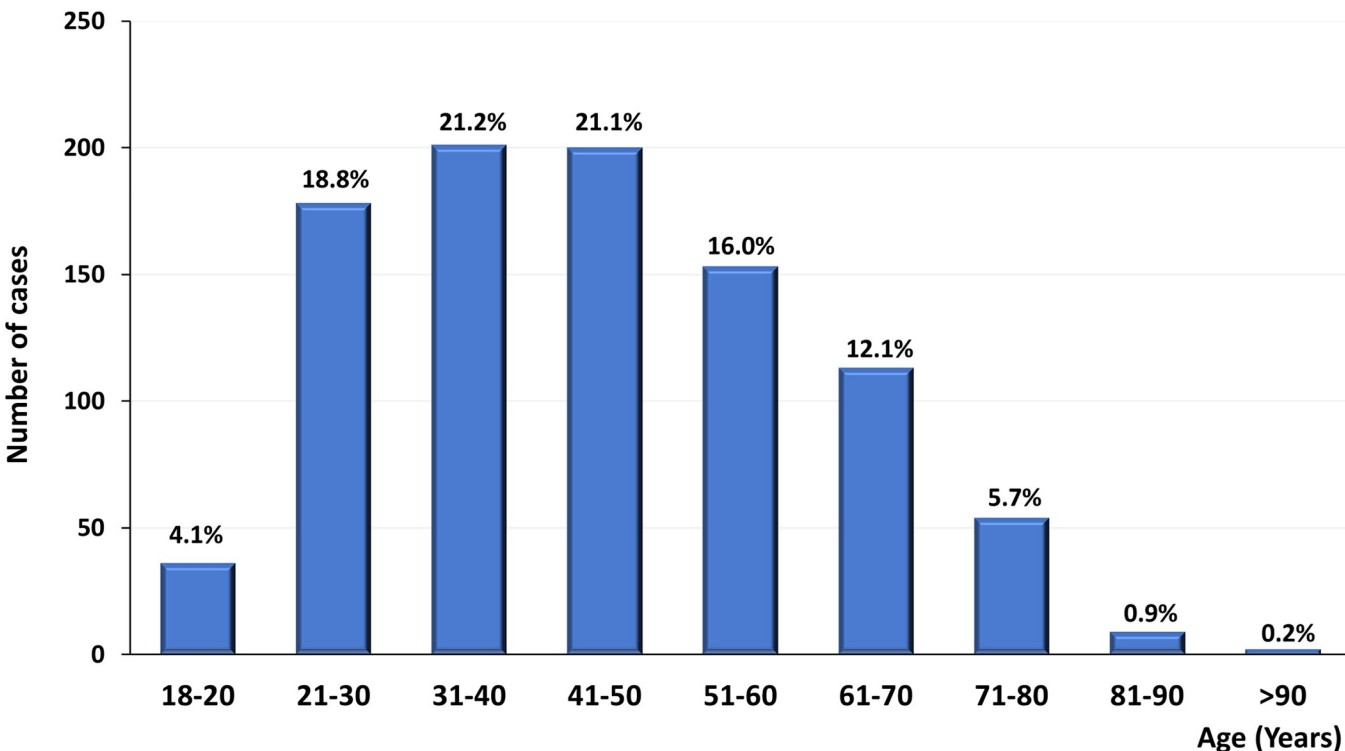

**Fig 2. Distribution of kidney biopsies based on different age groups.**

There was a slight predominance of females (52.6%). According to self-reported race, 44.9% were white, 45.4% were mulatto (mixed), 6.1% black, 1.3% had an Asian background, and 0.1% indigenous, reflecting the Brazilian ethnicity mix.

### 3.3. Clinical and laboratorial characteristics of the cohort

The most frequent clinical presentation in this cohort of adult patients was proteinuria (74.9% of the cases), presented as nephrotic syndrome in 41.4% and as non-nephrotic proteinuria in 33.5% (**Fig 3**). RPGN was the clinical manifestation of 6.7% of cases, whereas in additional 47.3% of the cases, a kidney biopsy was performed due to renal function impairment. Although hematuria was registered in 27.7% of the cases, it is noteworthy that only 7 patients presented hematuria as the exclusive indication for kidney biopsy, without other manifestations (Alport Syndrome/Thin basement membrane nephropathy n = 4, IgAN n = 1 and other glomerular diseases n = 2).

At kidney biopsy, 47.9% of patients had arterial hypertension. Only 15.2% of patients undergoing kidney biopsy in this registry were diabetic.

Age stratification of patients submitted to kidney biopsies according to their clinical indication is presented in **Table 1**. Nephrotic syndrome, rather than non-nephrotic proteinuria, was the major clinical indication of renal biopsy in patients 18–20 years (50.9%). On the other hand, hematuria was significantly less frequent in this age group. RPGN was more prominent for the ages of 21–30 and 31–40. The main indication of kidney biopsies for older adults was kidney dysfunction, associated or not with nephrotic syndrome/non-nephrotic proteinuria. As expected, hypertension and diabetes prevalence increased with age.

A family history of kidney disease was reported in 4% of patients; however, this information was not provided by centers in 47% of the overall sample.

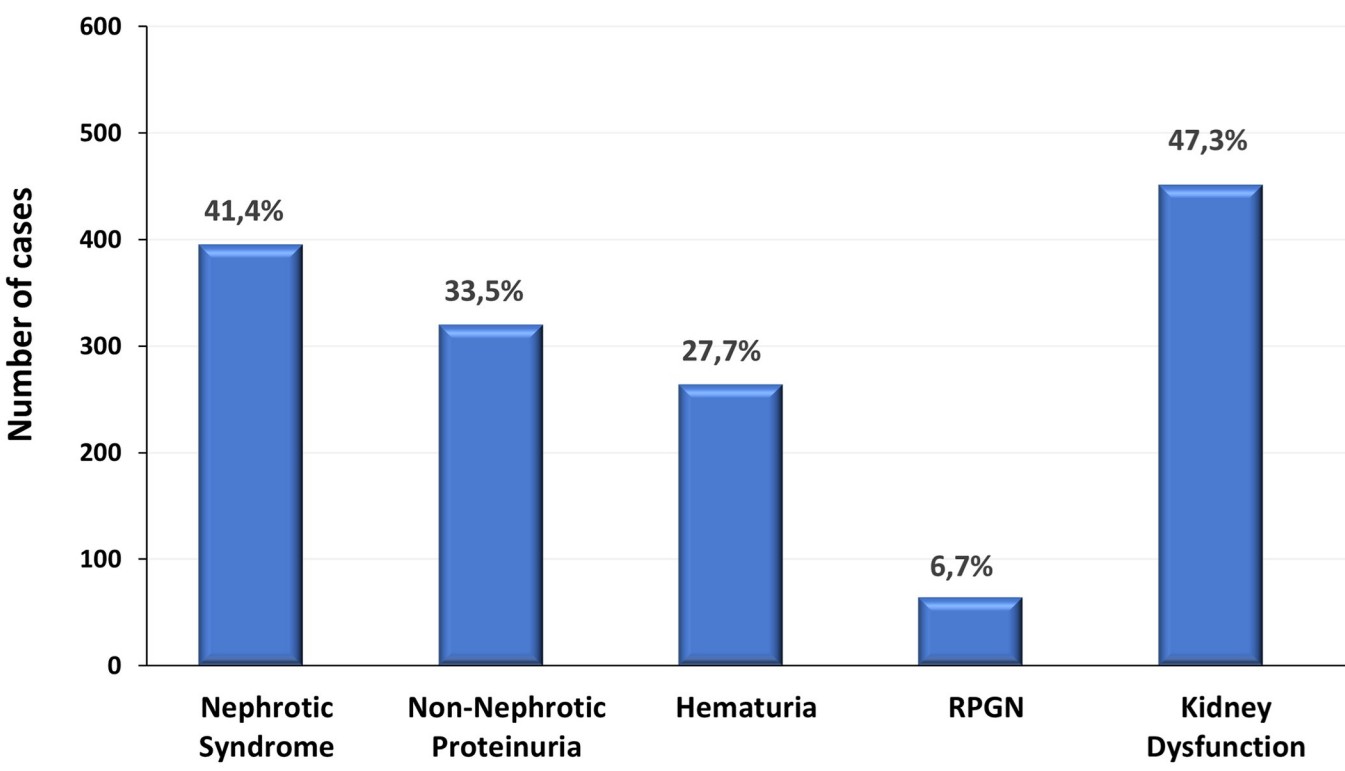

**Fig 3. Clinical presentation at kidney biopsy.**

**Table 1. Age stratification of patients submitted to kidney biopsies according to their clinical indication.**

| | Overall (n = 954) | | 18–20 (n = 39) | | 21–30 (n = 179) | | 31–40 (n = 202) | | 41–50 (n = 201) | | 51–60 (n = 153) | | 61–70 (n = 115) | | ≥71 y (n = 65) | |
|---|---|---|---|---|---|---|---|---|---|---|---|---|---|---|---|---|
| | n | (%) | n | (%) | n | (%) | n | (%) | n | (%) | n | (%) | n | (%) | n | (%) |
| Nephrotic Syndrome | 395 | (41.4%) | 23 | (59.0%)* | 61 | (34.1%) | 89 | (44.1%) | 85 | (42.3%) | 66 | (43.1%) | 42 | (36.5%) | 29 | (44.6%) |
| Non-nephrotic Proteinuria | 320 | (33.5%) | 12 | (6.7%)* | 66 | (36.9%) | 66 | (32.7%) | 66 | (32.8%) | 52 | (34.0%) | 41 | (35.7%) | 17 | (26.2%) |
| Hematuria | 264 | (27.7%) | 14 | (7.8%)* | 59 | (33.0%) | 48 | (23.8%) | 66 | (32.8%) | 46 | (30.1%) | 20 | (17.4%) | 11 | (16.9%) |
| RPGN | 63 | (6.6%) | 2 | (1.1%) | 16 | (8.9%)* | 17 | (8.4%)* | 7 | (3.5%) | 9 | (5.9%) | 9 | (7.8%) | 4 | (6.2%) |
| Kidney Dysfunction | 451 | (47.3%) | 7 | (17.9%) | 70 | (39.1%) | 87 | (43.1%) | 96 | (47.8%) | 89 | (58.2%) | 54 | (35.3%) | 48 | (73.8%)* |
| Hypertension | 457 | (47.9%) | 9 | (5.0%) | 59 | (33.0%) | 77 | (38.1%) | 90 | (44.8%) | 99 | (64.7%) | 78 | (67.8%) | 45 | (69.2%)* |
| Diabetes | 145 | (15.2%) | 0 | (0%) | 10 | (5.6%) | 17 | (8.4%) | 31 | (15.4%) | 33 | (21.6%) | 38 | (33.0%)* | 16 | (24.6%) |

*$p < 0.001$ vs other age intervals

At kidney biopsy, most patients (66.2%) presented eGFR$\leq$ 60ml/min/1,73m$^2$ (**Fig 4**). The mean serum creatinine was 2.7±2.9 mg/dL, with a mean eGFR of 49.5±36.9 ml/min/1.73m$^2$. Renal biopsy was performed in 17% of patients with eGFR< 15 ml/min/1.73m$^2$ (**Fig 4**).

### 3.4. Characteristics of kidney biopsy samples

The average number of glomeruli represented per biopsy was 19.8±12.9, with 81.2% of the cases (n = 775) with a representation of at least 10 glomeruli. Tubulointerstitial fibrosis and tubular atrophy (IFTA) was a common finding. The majority of kidney specimens (60.2%) had mild tubulointerstitial fibrosis/tubular atrophy on light microscopy analysis, while

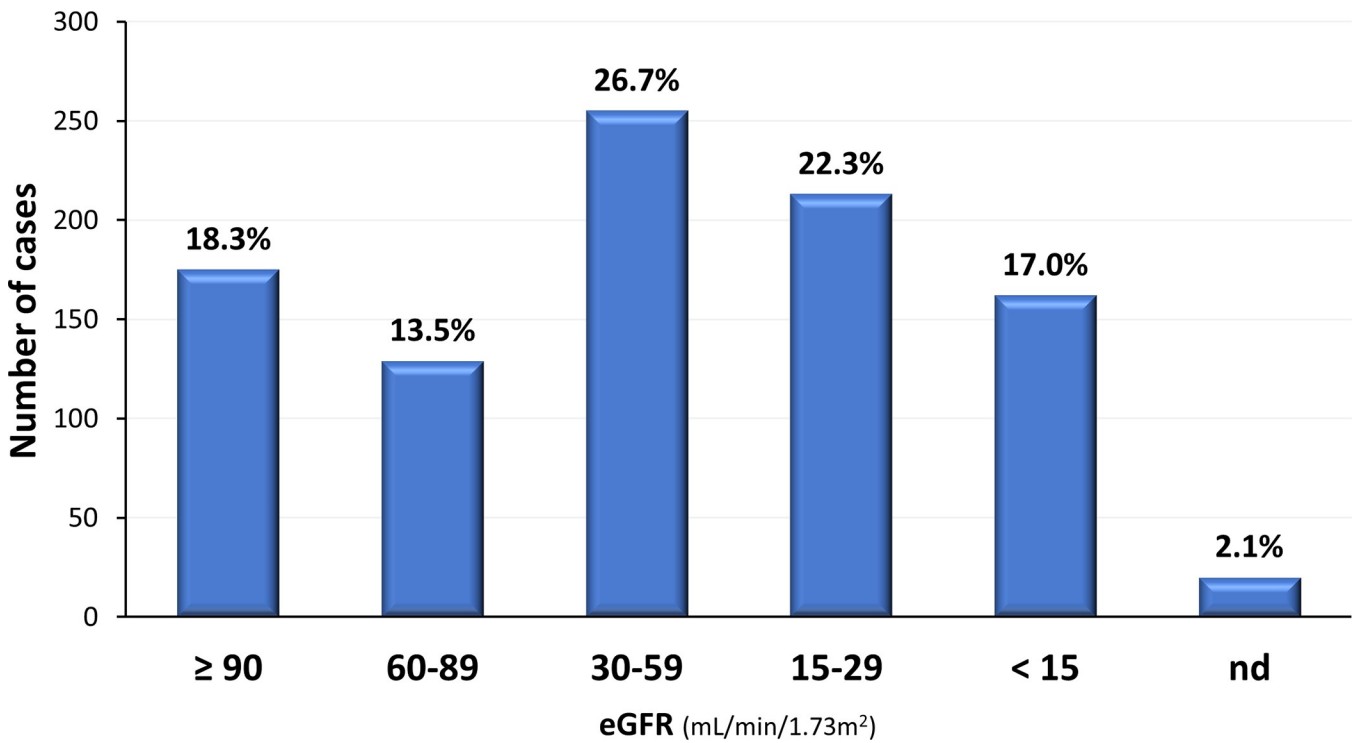

**Fig 4. eGFR and CKD stage at kidney biopsy.**

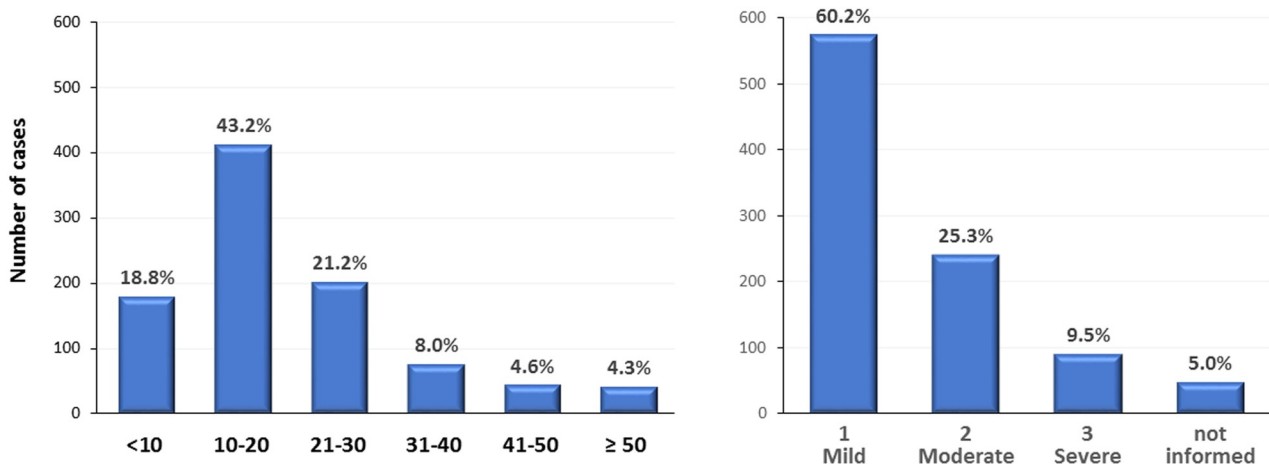

**Fig 5.** Characteristics of kidney biopsy: **A)** Number of glomeruli per biopsy; **B)** Degree of IFTA findings on kidney biopsies at kidney biopsy.

25.2% had moderate, and 9.5% had severe lesions; IFTA was not reported in 5% (**Fig 5**). Immunofluorescence or immunohistochemistry analyzes were performed on all samples, supporting the final diagnosis. Electronic microscopy was performed in 142 cases (15% of the cohort). Therefore, conditions such as Alport Syndrome, Thin Membrane Disease, Dense Deposit Disease, fibrillary and immunotactoid glomerulopathies, among others, were possibly misdiagnosed.

### 3.5. Diagnostic categories

The frequency of kidney biopsies grouped into the six main categories are presented in **Fig 6A**. The majority of biopsies were classified in the glomerular diseases group (74.8%). Diabetic nephropathy and monoclonal gammopathies were separately grouped and accounted for 6.4% and 4.6% of the cases, respectively. Tubulointerstitial diseases (5.5%) and vascular diseases (1%) were found in a minority of patients. Miscellaneous cases accounted for 1%. It is interesting to highlight the low percentage of biopsies in which an etiological diagnosis was not established (3.7%). The absolute numbers of all specific diagnoses included in this analysis are presented in **Fig 6B**.

Table 2 shows demographic, clinical and laboratory characteristics of the kidney biopsies grouped in main categories. For glomerular and inherited kidney diseases, kidney biopsies were performed in the thirties and forties decades of life, whereas for monoclonal gammopathies predominantly in the fifties and sixties decades. There was no racial disparity among the groups. As expected, proteinuria was the main laboratory finding (77.9%) in the glomerular diseases group, presenting as nephrotic syndrome or non-nephrotic proteinuria. Hematuria was less frequent (30.4% of the cases). In the diabetic nephropathy and monoclonal gammopathies groups, nephrotic proteinuria was a prominent feature as well as renal dysfunction. Inherited kidney diseases presented mainly as non-nephrotic proteinuria, hematuria, and renal dysfunction was rarely presented at kidney biopsy.

Tubulointerstitial diseases mainly manifested as renal dysfunction, rarely with nephrotic syndrome. Similarly, kidney dysfunction was also the most frequent manifestation of vascular diseases submitted to a kidney biopsy. Interestingly, 26.9% of the patients with vascular diseases diagnosed on kidney biopsies presented nephrotic syndrome.

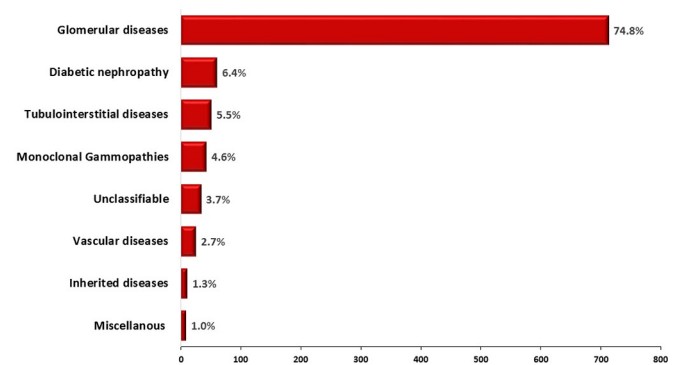

A

| | |
|---|---|
| Glomerular diseases | 74.8% |
| Diabetic nephropathy | 6.4% |
| Tubulointerstitial diseases | 5.5% |
| Monoclonal Gammopathies | 4.6% |
| Unclassifiable | 3.7% |
| Vascular diseases | 2.7% |
| Inherited diseases | 1.3% |
| Miscellanous | 1.0% |

B

| Glomerular Diseases | 714 | 74.8% |
|---|---|---|
| Lupus nephritis | 216 | 22.6% |
| IgAN | 124 | 13.0% |
| FSGS | 116 | 12.2% |
| Membranous Nephropathy | 80 | 8.4% |
| Minimal Change Disease | 40 | 4.2% |
| Vasculitis - Pauci Imune GN | 32 | 3.4% |
| Collapsing Glomerulopathy | 24 | 2.5% |
| IC-MPGN | 20 | 2.1% |
| TMA | 16 | 1.7% |
| Acute Post-Infectious GN | 11 | 1.2% |
| C3-Glomerulopathy | 10 | 1.0% |
| Crescentic GN - Immune Complex Mediated | 8 | 0.8% |
| HIV-Related Nephropathy* | 6 | 0.6% |
| Crescentic GN - anti-GBM GN | 5 | 0.5% |
| Mesangial Proliferative GN | 4 | 0.4% |
| Cryoglobulinemic GN | 2 | 0.2% |

| Diabetic Nephropathy | 61 | 6.4% |
|---|---|---|

| Tubulo-Interstitial Diseases | 52 | 5.5% |
|---|---|---|
| Acute Interstitial Nephritis | 40 | 4.2% |
| ATN | 11 | 1.2% |
| IgG4-Related Kidney Disease | 1 | 0.1% |

| Monoclonal Gammopathy | 44 | 4.6% |
|---|---|---|
| Amyloidosis | 12 | 1.3% |
| Cast Nephropathy | 9 | 0.9% |
| MIDD | 8 | 0.8% |
| Light Chain Proximal Tubulopathy | 6 | 0.6% |
| Other Gammopathies | 6 | 0.6% |
| Fibrillary Glomerulopathy | 1 | 0.1% |
| Immunotactoid Glomerulopathy | 1 | 0.1% |
| PGNMID | 1 | 0.1% |

| Vascular Diseases | 26 | 2.7% |
|---|---|---|
| Hypertensive Nephrosclerosis | 25 | 2.6% |
| Systemic Sclerosis | 1 | 0.1% |

| Inherited Kidney Diseases | 12 | 1.3% |
|---|---|---|
| Collagen IV disorders | 9 | 0.9% |
| LCAT Deficiency | 2 | 0.2% |
| Fabry´s Disease | 1 | 0.1% |

| Miscellaneous | 10 | 1.0% |
|---|---|---|

| Unclassifiable | 35 | 3.7% |
|---|---|---|

**Fig 6. Histological diagnosis of all kidney biopsies of this cohort classified into six main categories. 6A)** Frequencies of kidney biopsies diagnosis grouped into the six main categories. **6B)** Detailed numbers and percentages of all specific diagnoses, separated according to the categories.

## 3.6. Glomerular involvement in kidney biopsies in the Brazilian cohort

In order to more specifically evaluate glomerular diseases in this cohort, all kidney biopsies with glomerular involvement were subgrouped (**S2B Table**). A total of 810 biopsies were reclassified as glomerular diseases (**Fig 7**), representing a Brazilian incidence of glomerular diseases of 0.38/100,000 inhabitants/year.

Lupus nephritis was the most prevalent glomerular disease diagnosed in kidney biopsies of the BKBR (22.6%). IgAN (13%) predominated in the biopsies performed in Brazil in 2021 and included in this registry, followed by FSGS (12.2%) and membranous nephropathy (8.4%). Diabetic nephropathy as the main glomerular disease was observed in 6.4% of cases. The frequency of other glomerular diseases is presented in **Fig 7**.

**Table 2. Demographic, clinical and laboratory characteristics of the kidney biopsies grouped in main categories.**

| | All Cases | Glomerular diseases | Diabetic Nephropathy | Monoclonal Gammopathies | Inherited kidney diseases | Tubulointersti-tial diseases | Vascular diseases | Miscellaneous and Unclassfied |
|---|---|---|---|---|---|---|---|---|
| **Number (%)** | 954 (100%) | 714 (74.8%) | 61 (6.4%) | 44 (4.6%) | 12 (1.3%) | 52 (5.5%) | 26 (2.7%) | 45 (4.7%) |
| **Age** (y) | 44.7 ± 15.9 | 47 ± 2.9 | 50.9 ± 14.1 | 61.3 ± 2.0 | 39.8 ± 9.0 | 53.6 ± 14.0 | 50.8 ± 1.8 | 45.6 ± 14.8 |
| **Gender** | | | | | | | | |
| Male | 452 (47.4%) | 342 (45.9%) | 29 (47.5% | 25 (56.8%) | 6 (50%) | 38 (51.9%) | 14 (53.8%) | 23 (51.1%) |
| Female | 502 (52.6%) | 396 (54.1%) | 32 (52.5%) | 19 (43.2%) | 6 (50%) | 34 (48.1%) | 12 (46.2%) | 22 (48.9%) |
| **Race** | | | | | | | | |
| White | 428 (44.9%) | 327 (44.4%) | 25 (41%) | 19 (36.5%) | 5 (41.7%) | 35 (47.9%) | 11 (42.3%) | 25 (55.6%) |
| Black | 58 (6.1%) | 45 (6.1%) | 3 (4.9%) | 1 (1.9%) | 0 (0%) | 4 (5.5%) | 2 (7.7%) | 4 (8.9%) |
| Mulatto (mixed) | 455 (47.7%) | 357 (48.4%) | 32 (52.5%) | 24 (46.2%) | 7 (58.3%) | 32 (43.8%) | 12 (46.2%) | 15 (33.3%) |
| Yellow (Asian) | 12 (1.3%) | 7 (0.9%) | 1 (1.6%) | 0 (0%) | 0 (0%) | 2 (2.7%) | 1 (3.8%) | 1 (2.2%) |
| Indigenous | 1 (0.1%) | 1 (0.1%) | 0 (0%) | 0 (0%) | 0 (0%) | 0 (0%) | 0 (0%) | 0 (0%) |
| **Clinical characteristics** | | | | | | | | |
| Nephrotic syndrome | 395 (41.4%) | 312 (43.7%) | 35 (57.4%) | 21 (47.7%) | 3 (25%) | 1 (1.9%)* | 7 (26.9%)# | 16 (35.6%) |
| Non-nephrotic proteinuria | 320 (33.5%) | 244 (34.2%) | 15 (24.6%) | 8 (18.2%) | 7 (58.3%) | 18 (34.6%) | 11 (42.3%) | 17 (37.8%) |
| Hematuria | 264 (27.7%) | 217 (30.4%) | 8 (13.1%) | 6 (13.6%) | 7 (58.3%) | 10 (19.2%) | 4 (15.4%) | 12 (26.7%) |
| RPGN | 63 (6.6%) | 56 (7.8%) | 0 (0%) | 1 (2.3%) | 0 (0%) | 4 (7.7%) | 1 (3.8%) | 1 (2.2%) |
| Kidney dysfunction | 451 (47.3%) | 295 (41.3%) | 41 (67.2%) | 31 (70.5) | 1 (8.3%) | 45 (86.5%) | 17 (65.4%) | 21 (46.7%) |
| **Serum creatinine** at kidney Bx (mg/dL) | 2.7 ± 2.9 | 3.6 ± 2.0 | 3.2 ± 3.2 | 4.2±0.9 | 1.0 ± 0.3 | 3.0 ± 2.0 | 2.5 ± 3.6 | 2.7 ± 1.3 |

*p <0.001 *vs* other group categories

#p <0.005 *vs* other histologic diagnosis

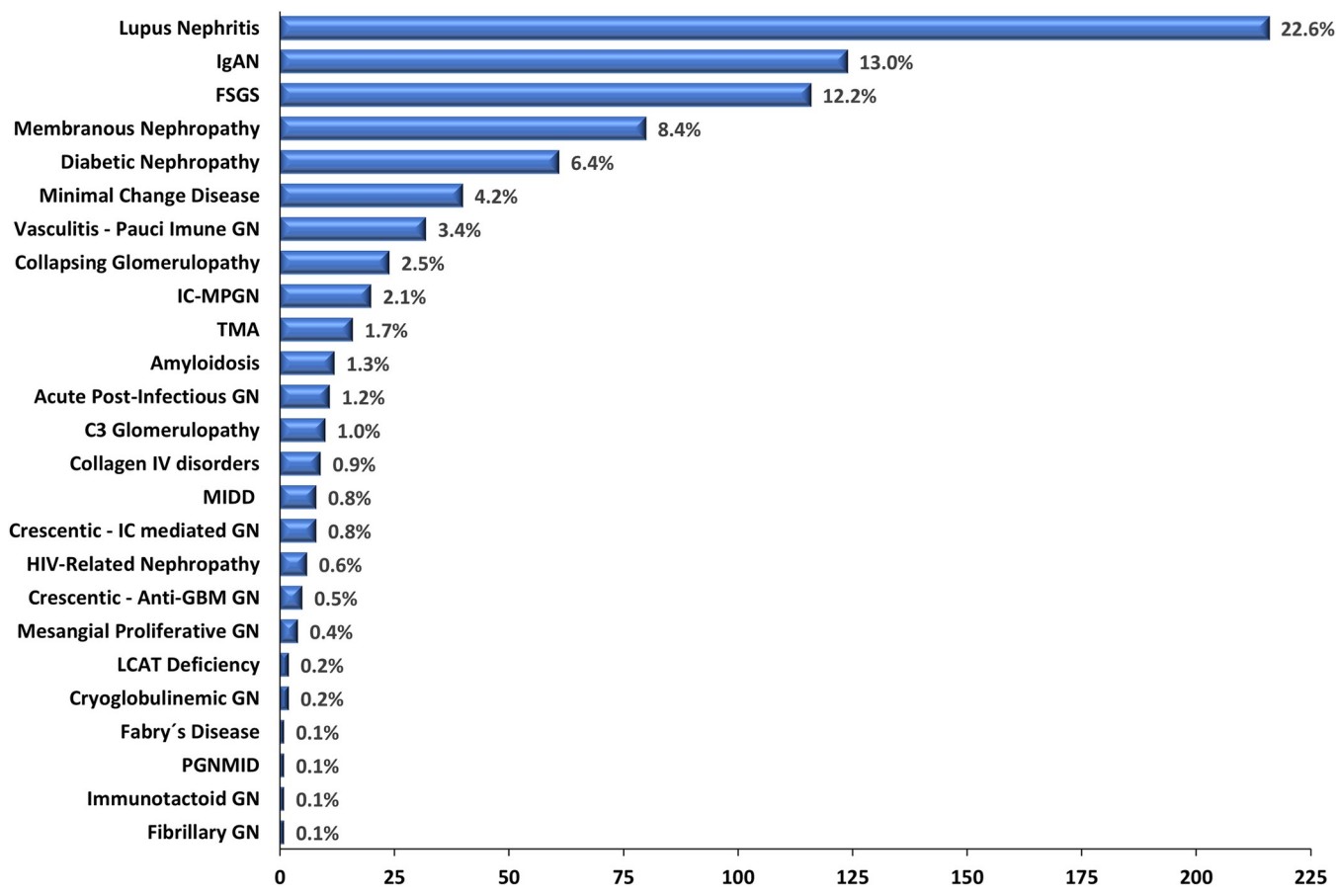

**Fig 7. Frequencies of diseases with glomerular involvement in kidney biopsies of the BKBR in 2021.**

**3.6.1. Stratification of glomerular diseases according to age.** Lupus nephritis was the most frequent histologic diagnosis in the age range from 21–50 years. IgAN and FSGS were mainly diagnosed in the age groups of 21–60 years, whereas Minimal Change Disease manifested in the 21-40s, and Collapsing Glomerulopathy in young adults. Membranous nephropathy and diabetic nephropathy were more prevalent in older stratification, aged > 40 years (**Fig 8**). Similarly, diseases classified as monoclonal gammopathies, such as amyloidosis, MIDD, cryoglobulinemic GN, PGNMID, immunotactoid and fibrillary GN, as well as vasculitis (pauci immune GN) were diagnosed in older patients. On the other hand, collagen IV disorders and LCAT-Deficiency were diagnosed in younger patients (Table 3). A specific analysis in the elderly population (age ≥ 65 y) is presented in **S3 Table**.

**3.6.2. Stratification of glomerular diseases according to gender.** As expected, a marked preponderance of lupus nephritis was observed in females versus males (5.8:1), whereas IgAN, MN, collapsing glomerulopathy and acute post-infectious glomerulonephritis were more prevalent among males compared with females (1.8:1, 1.6:1, 5:1 and 4.5:1, respectively).

**3.6.3. Clinical characteristics at kidney biopsy according to the diagnosis of glomerular diseases.** At kidney biopsy, lupus nephritis patients had a clinical presentation consisting of nephrotic and non-nephrotic proteinuria, hematuria and kidney dysfunction. However, nephrotic syndrome was not so frequent in IgAN cases. In contrast, nephrotic syndrome was the most prominent clinical manifestation in minimal change disease and membranous nephropathy (**Fig 9**). In FSGS, collapsing glomerulopathy and diabetic nephropathy, as

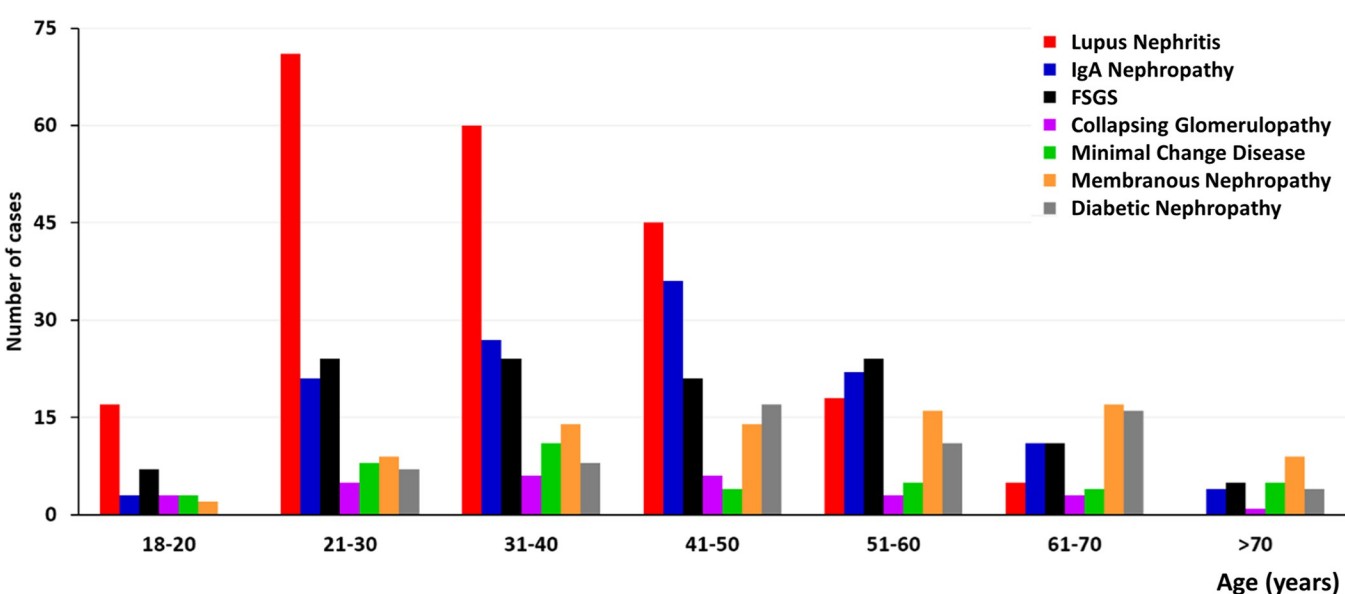

**Fig 8. Stratification of glomerular diseases according to age.**

expected, nephrotic syndrome was also an important clinical feature, however, at kidney biopsy, patients also presented kidney dysfunction.

In cases of monoclonal gammopathies with glomerular involvement (such as amyloidosis, MIDD, PGNMID, fibrillary and immunotactoid glomerulopathies; n = 23), 69.6% had nephrotic syndrome, 17.4% non-nephrotic proteinuria, 13% hematuria, and 69.6% presented kidney dysfunction.

**3.6.4. eGFR at kidney biopsy according to the diagnosis of glomerular diseases.** In diseases such as lupus nephritis, minimal change disease and membranous nephropathy the majority of kidney biopsy was performed in patients with preserved renal function (eGFR > 90ml/min/1.73m$^2$). In patients with IgAN and FSGS the majority of the diagnosis was done in CKD stage 3. On the other hand, the diagnosis of diabetic nephropathy in this cohort was performed at CKD stage 4 and 5 (**Fig 10**).

### 3.7. Lupus nephritis

Lupus nephritis was the most common form of glomerular disease in the present registry. The analysis of histological classes showed that class IV (diffuse proliferative) was the most frequent (45%), followed by the association of classes III+V or IV+V (22%) and class V (15%). It is important to highlight the presence of cases of podocytopathy in 1% of cases.

### 3.8. FSGS

The 116 FSGS cases were sub grouped according to the pathologic variants based on the Columbia FSGS classification [7] (Table 4). Not-otherwise specified (NOS) was the most frequent subtype, diagnosed in 55.2%, followed by Tip lesion (11.2%), and Perihilar (4.3%). A subclass of FSGS was not informed in 29.3%. TIP lesion occurred in earlier ages (37.1±12.4), whereas perihilar subtype in older ages (56.6±20.1). The majority of cases diagnosed as TIP lesion presented nephrotic syndrome (76.9%). History of diabetes was present in 20% of perihilar cases and hypertension in 80%.

Table 3. Distribution of glomerular diseases subtypes according to gender and self-declared race.

| | n | % | age mean±SD | Male n (%) | Female n (%) | | B n (%) | | P n (%) | | M n (%) | | A n (%) | | I n (%) | |
|---|---|---|---|---|---|---|---|---|---|---|---|---|---|---|---|---|
| Lupus nephritis | 216 | 22.6% | 35±11.6 | 32 (14.8%) | 184 | (85.2%) | 98 | (45.4%) | 16 | (7.4%) | 102 | (47.2%) | 0 | 0% | 0 | 0% |
| IgAN | 124 | 13.0% | 43.7±13.0 | 81 (65.3%)* | 43 | (34.7%) | 48 | (38.7%) | 2 | (1.6%) | 72 | (58.1%) | 2 | 1.6% | 0 | 0% |
| FSGS | 116 | 12.2% | 42.8±15.3 | 65 (56%) | 51 | (44.0%) | 58 | (50%) | 8 | (6.9%) | 48 | (41.4%) | 2 | 1.7% | 0 | 0% |
| Membranous Nephropathy | 80 | 8.4% | 51.1±17.1 | 50 (62.5%)* | 30 | (37.5%) | 31 | (38.8%) | 9 | (11.3%) | 39 | (48.8%) | 1 | 1.3% | 0 | 0% |
| Diabetic Nephropathy | 61 | 6.4% | 50.9±14.1 | 29 (47.5%) | 32 | (52.5%) | 25 | (41.0%) | 3 | (4.9%) | 32 | (52.5%) | 1 | 1.6% | 0 | 0% |
| Minimal Change Disease | 40 | 4.2% | 44.5±18.4 | 18 (45%) | 22 | (55.0%) | 17 | (42.5%) | 0 | (0%) | 22 | (55.0%) | 1 | 2.5% | 0 | 0% |
| Vasculitis—Pauci Immune GN | 32 | 3.4% | 51±15.8 | 16 (50%) | 16 | (50%) | 19 | (59.4%) | 0 | (0%) | 11 | (34.4%) | 1 | 3.1% | 1 | 2% |
| Collapsing Glomerulopathy | 24 | 2.5% | 41.0±15.3 | 20 (83.3%)* | 4 | (16.7%) | 11 | (45.8%) | 5 | (20.8%) | 8 | (33.3%) | 0 | 0% | 0 | 0% |
| IC-MPGN | 20 | 2.1% | 50.0±16.2 | 9 (45.0%) | 11 | (55.0%) | 10 | (50%) | 1 | (2.0%) | 9 | (18.4) | 0 | 0% | 0 | 0% |
| TMA | 16 | 1.7% | 49.2±17.8 | 9 (56.3%) | 7 | (43.8%) | 7 | (43.8%) | 3 | (18.8%) | 6 | (37.5%) | 0 | 0% | 0 | 0% |
| Amyloidosis | 12 | 1.3% | 58.8±12.1 | 6 (50%) | 6 | (50.0%) | 5 | (41.7%) | 0 | (0%) | 7 | (58.3%) | 0 | 0% | 0 | 0% |
| Acute Post-Infectious GN | 11 | 1.2% | 39.2±13.7 | 9 (81.8%) | 2 | (18.2%) | 5 | (45.5%) | 0 | (0%) | 6 | (54.5%) | 0 | 0% | 0 | 0% |
| C3 Glomerulopathy | 10 | 1.0% | 45.1±16.9 | 7 (70.0%) | 3 | (30%) | 4 | (40%) | 0 | (0%) | 6 | (12.5%) | 0 | 0% | 0 | 0% |
| Collagen IV disorders | 9 | 0.9% | 36.3±12.7 | 4 (44.4%) | 5 | (55.6%) | 2 | (22.2%) | 0 | (0%) | 7 | (77.8%) | 0 | 0% | 0 | 0% |
| MIDD | 8 | 0.8% | 57.3±15.3 | 6 (75%) | 2 | (25.0% | 4 | (50%) | 0 | (0%) | 4 | (50%) | 0 | 0% | 0 | 0% |
| Crescentic—IC Mediated | 8 | 0.8% | 44.4±20.0 | 2 (25%) | 6 | (75.0%) | 1 | (12.5%) | 0 | (0%) | 7 | (87.5%) | 0 | 0% | 0 | 0% |
| HIV-Related Nephropathy | 6 | 0.6% | 52.3±14.4 | 4 (66.7%) | 2 | (33.3%) | 3 | (50%) | 1 | (16.7%) | 2 | (33.3% | 0 | 0% | 0 | 0% |
| Crescentic—Anti-GBM GN | 5 | 0.5% | 49±17.2 | 3 (60%) | 2 | (40%) | 2 | (40%) | 0 | (0%) | 3 | (60%) | 0 | 0% | 0 | 0% |
| Mesangial Proliferative GN | 4 | 0.4% | 48.5±19.2 | 2 (50%) | 2 | (50%) | 2 | (50%) | 0 | (0%) | 2 | (50%) | 0 | 0% | 0 | 0% |
| LCAT Deficiency | 2 | 0.2 | 33±12.7 | 2 (100%) | 0 | (0%) | 2 | (100%) | 0 | (0%) | 0 | (0%) | 0 | 0% | 0 | 0% |
| Cryoglobulinemic GN | 2 | 0.2% | 65.5±9.2 | 1 (50%) | 1 | (50%) | 1 | (50%) | 0 | (0%) | 1 | (50%) | 0 | 0% | 0 | 0% |
| Fabry Disease | 1 | 0.1% | 50 | 0 (0.0%) | 1 | (100%) | 1 | (100%) | 0 | (0%) | 0 | (0%) | 0 | 0% | 0 | 0% |
| PGNMID | 1 | 0.1% | 65 | 1 (100%) | 0 | (0%) | 1 | (100%) | 0 | (0%) | 0 | (0%) | 0 | 0% | 0 | 0% |
| Immunotactoid GN | ʻ1 | 0.1% | 51 | 1 (100%) | 0 | (0%) | 0 | (0%) | 0 | (0%) | 1 | (100%) | 0 | 0% | 0 | 0% |
| Fibrillary GN | 1 | 0.1% | 73 | 0 (0%) | 1 | (100%) | 0 | (0%) | 0 | (0%) | 1 | (100%) | 0 | 0% | 0 | 0% |
| OVERALL | 810 | 84.9% | 49.2±2.8 | 377 | 434 | | 357 | | 48 | | 396 | | 8 | | 1 | |

(*p <0.001 vs other gender

#p <0.019 vs other gender)

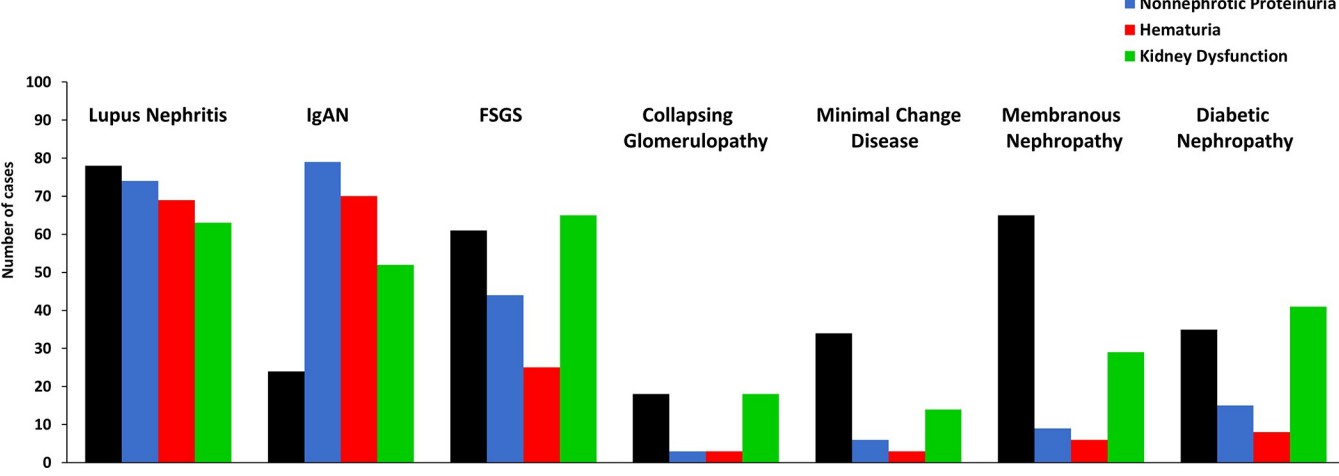

Fig 9. Clinical presentation at kidney biopsy according to the diagnosis of glomerular diseases.

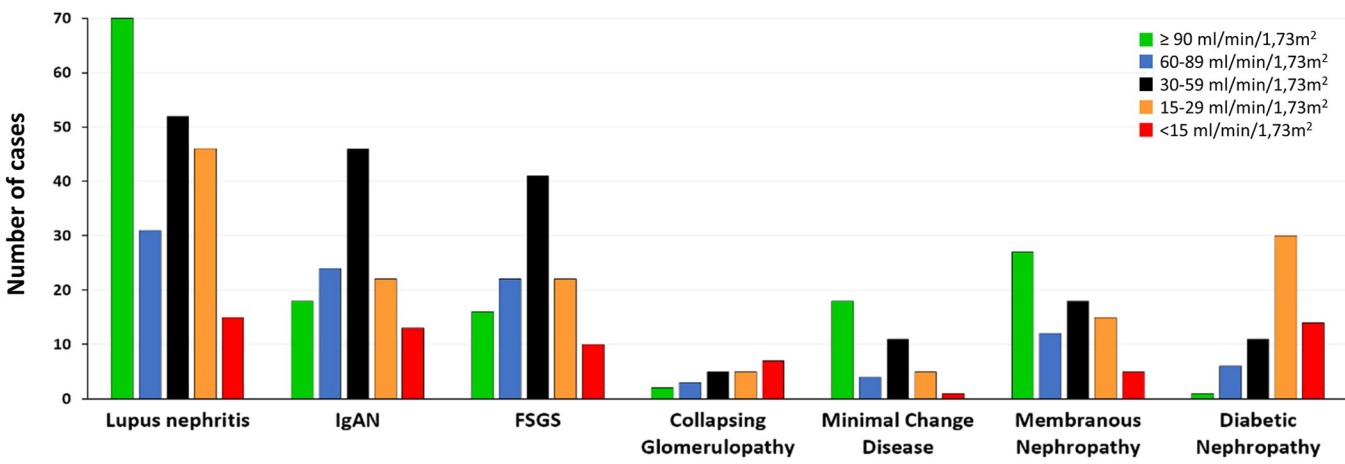

**Fig 10. eGFR at kidney biopsy according to the diagnosis of glomerular diseases.**

### 3.9. Diabetic nephropathy

Of 145 diabetic patients that underwent a kidney biopsy, 81 (56%) had histological evidence of diabetic nephropathy (8.5% of the cohort; Fig 11). From these 81 biopsies, diabetic nephropathy was the main diagnosis of the kidney disease in 61 cases (in 12, histological evidence of hypertensive nephrosclerosis was also present, and 2 cases also presented thrombotic microangiopathy). The other 20 diabetic patients with histological features of diabetic nephropathy, however, presented other predominant kidney diseases diagnosed by histological and clinical features.

Of 145 diabetic patients, 64 (44%) had no evidence of diabetic nephropathy on kidney biopsy. These patients presented the following diagnosis: Lupus nephritis n = 9; Acute Interstitial Nephritis n = 7; Minimal Change Disease n = 6; Membranous Nephropathy n = 6; Crescentic GN n = 4 (3 Vasculitis and 1 Immune Complex Mediated Crescentic GN); Hypertensive nephrosclerosis n = 4 (1 Malignant); Unclassifiable n = 4; Collapsing Glomerulopathy n = 3; FSGS n = 2; IgAN n = 2; C3-Glomerulopathy n = 2; Thrombotic microangiopathy n = 2; ATN n = 2; Miscellaneous n = 2; Amyloidosis n = 1; Cast Nephropathy n = 1; IgG4-Related Kidney Disease n = 1; Fibrillary Glomerulopathy n = 1; Mesangial Proliferative GN n = 1;

**Table 4. Baseline characteristics of patients with FSGS at the kidney biopsy.**

| | | Age (at the Bx) | Gender (Male) | Race (White) | Serum creatinine (mg/dL) | Nephrotic Syndrome | Non-Nephrotic proteinuria | Hematuria | Renal Dysfunction | Diabetes | Hypertension |
|---|---|---|---|---|---|---|---|---|---|---|---|
| **Overall** | 116 | 42. 8 ± 15.3 | 65 (56 0%) | 58 (50%) | 2.1±1.5 | 61 (52.6%) | 44 (37.9%) | 25 (21.6%) | 72 (62.15) | 4 (3.4%) | 59 (50.9%) |
| **NOS** | 64 (55.2%) | 44 0±14.8 | 38 (59.4%) | 34 (53.1%) | 2.2±1.6 | 33 (51.6%) | 21 (32.8%) | 11 (17.2%) | 45 (70.3%) | 1 (1.6%) | 37 (57.8%) |
| **TIP lesion** | 13 (11.2%) | 37.1±12.4 | 8 (61.5%) | 6 (46.2%) | 2.0±0.9 | 10 (76.9%) | 3 (23.1%) | 2 (15.4%) | 3 (4.7%) | 0 | 3 (23.1%) |
| **Perihilar** | 5 (4.3%) | 56.6±20.1 | 3 (60.0%) | 3 (60.0%) | 1.8±1.5 | 3 (60.0%) | 2 (40.0%) | 0 | 5 (7.8%) | 1 (20.0%) | 4 (80.0%) |
| **Not informed** | 34 (29.3%) | 40.6±15.6 | 16 (45.7%) | 15 (42.9%) | 2.0±1.6 | 15 (42.9%) | 18 (51.4%) | 12 (34.3%) | 19 (29.7%) | 2 (5.7%) | 15 (42.9%) |

NOS: not otherwise specified

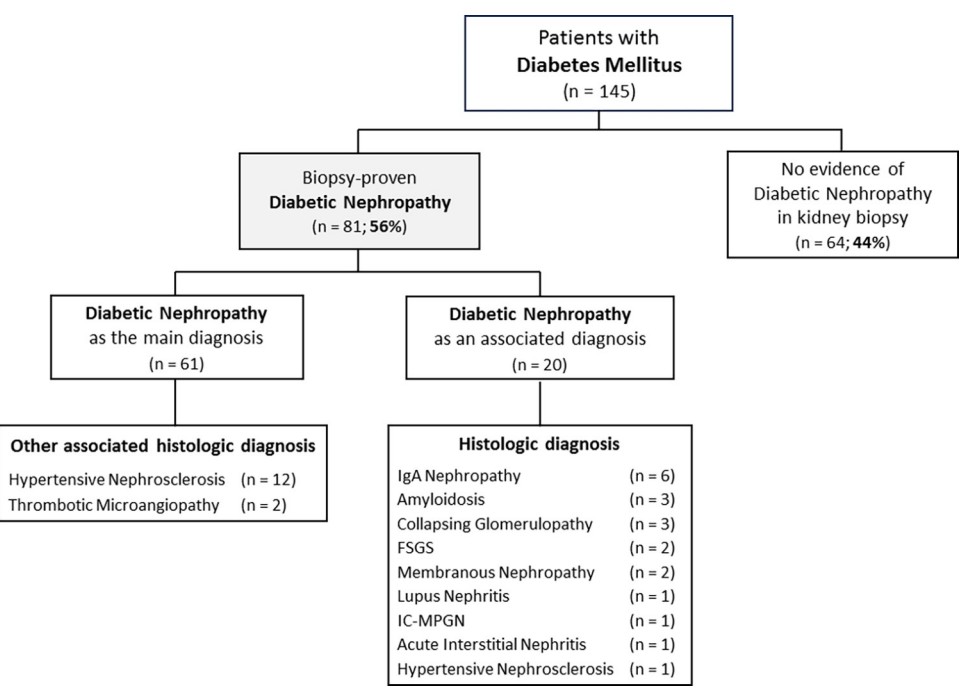

**Fig 11. Kidney biopsy findings in patients with diabetes mellitus.**

IC-MPGN n = 1 (Hepatitis C); MPGN with negative IF n = 1; Systemic Sclerosis n = 1; and Thin Basement Membrane Nephropathy n = 1. Demographic, clinical and laboratory characteristics of the 145 patients with diabetes mellitus are presented in **S4 Table**.

### 3.10. Thrombotic microangiopathy

There were 16 cases of primary TMA and 23 of secondary TMA; from these 23, 9 cases were associated with lupus nephritis, 4 with diabetic nephropathy, 3 with IgAN, 3 with collapsing glomerulopathy, 2 cases with malignant nephrosclerosis, and 2 cases of acute interstitial nephritis.

### 3.11. Virus-associated kidney diseases

There were 20 cases of nephropathies associated with COVID: 7 FSGS, 4 diabetic nephropathy, 2 immune complex glomerulonephritis, 3 IgAN, 2 TMA and 1 acute interstitial nephritis, and 1 malignant nephrosclerosis.

There were 6 cases of HIV-Related Nephropathy, two cases presented nephrotic syndrome (HIVAN), 2 cases were HIVICK (1 membranous nephropathy and 1 immune complex glomerulonephritis) and 2 cases of acute interstitial nephritis.

## 4. Discussion

Although Brazil has established registries for dialysis and transplantation revealing more than 150,000 patients on dialysis (the fourth largest dialysis population worldwide), and the largest public kidney transplant program worldwide with more than 9,000 kidney transplants performed per year, studies on the epidemiology of glomerular diseases in Brazil are scarce. The establishment of a nationwide BKBR represents an important tool to better characterize the epidemiology of glomerular diseases in Brazil. The development of a web-based registry

platform with technical characteristics to guarantee safety and efficiency enabled a rapid input of data relative to patient demographics, clinical data and the diagnosis obtained by native kidney biopsies. The first year of operation of this registry was 2021, registering 1012 kidney biopsies. A cohort of 954 cases was analyzed and reported in this manuscript.

All regions of Brazil, a country with continental dimensions with great geographic, social, and economic diversity, were represented, although the highest number of cases registered was in the Southeast region, reflecting the major nephrology activity with the concentration of dialysis and transplant centers in this region.

Notwithstanding the age peak incidence in the range of 21–50 years, in agreement with previous reports [4, 8–10], it is interesting to observe a higher number of biopsies performed in older patients, as also reported by Cunningham *et al* [11]; possibly reflecting the overall demographic changes with the increase in life expectancy, and non-communicable diseases such as diabetes and hypertension becoming more prevalent, with a consequent increase in the prevalence of CKD in the elderly [12]. In fact, nephrotic syndrome and kidney function impairment represent the most important referral for kidney biopsy in this population, while this frequency in international registries varied from 10.3% to 25.8% [13–15].

Unlike countries where Caucasians are predominant [16], the Brazilian population has one of the most mixed genetic backgrounds as a result of multi-ethnic ancestries, particularly of European, American Indigenous, and African descents. There is great variability in skin pigmentation, with genomic ancestry prediction being unreliable [17, 18]. According to the most recent Brazilian census [19] based on self-reported race, 43.5% declare themselves as white, 10.2% black, 44.2% mulattos, 1.7% indigenous and 0.4% yellow. The representation of different racial groups in this cohort mirrored the Brazilian distribution, with no marked specific correlation with kidney diseases. This even holds for pathologies such as FSGS and lupus nephritis, which might have a higher frequency in blacks or IgAN in the Southwest region of Brazil, particularly in São Paulo, with a large population of Asian descendants, similar to reported by Sim *et al.* [20].

Kidney biopsy indication is mainly driven by proteinuria and kidney function impairment, rather than hematuria. Proteinuria was the most frequent clinical manifestation at kidney biopsy, similar to reported in other registries [3, 4, 21–24] However, it is of note that almost half of patients submitted to a kidney biopsy in this registry presented kidney function impairment. Kidney dysfunction was particularly prominent in older adult patients ($\geq$ 71y), possibly explaining the indication of a kidney biopsy in this group of older patients. Kidney dysfunction in patients $\geq$ 65y was a prominent clinical feature particularly in acute interstitial nephritis, monoclonal gammopathies, and TMA.

The majority of MCD, membranous nephropathy, lupus nephritis, and inherited diseases are submitted to a kidney biopsy in CKD stage 1. However, most patients in the BKBR presented eGFR <60 ml/min/1.73m$^2$ at the time of the kidney biopsy in CKD stages 3, 4 and remarkably in stage 5. These results may reflect difficulties in accessing health services, particularly in 2021 during the coronavirus pandemic. In addition, more aggressive kidney diseases or diseases that progress with rapid kidney dysfunction, such as collapsing glomerulopathy, crescentic glomerulonephritis, monoclonal gammopathies, tubulointerstitial diseases, malignant nephrosclerosis, and a proportion of lupus nephritis cases may account for these results. The results that 60% of the biopsies presented a mild degree of IFTA suggests that the predominance of patients with eGFR <60 ml/min/1.73m$^2$ at the kidney biopsy is not related to histologic markers of chronicity [25].

Classifying the kidney biopsies into 6 main categories was interesting to have a better overview on the frequencies and characteristics of different groups of kidney diseases in this Brazilian cohort, which is similar to other reports [10] In addition to the expected results of

glomerular diseases as the most frequent diagnosis, it is important to highlight the prevalence of diabetic nephropathy confirmed by biopsy, as recently highlighted [16, 26] and also the diagnosis of monoclonal gammopathies, an entity better recognized in recent years. The low frequency of inherited kidney diseases in this registry is also noteworthy, probably associated with limited electron microscopy analysis only being performed in 15% of the overall biopsies. In this regard, this registry also highlights the importance of strengthening analyses which enable diagnosis, such as more broadly implemented electron microscopy, histologic markers, and genetic testing. Despite the relative low frequency of inherited diseases in this cohort, it represents an increase of 70% compared to a previous Brazilian report [4].

The prevalence of glomerular diseases varies among countries and even among regions of the same country. According to the United States Renal Data System [27] glomerulonephritis also ranks as the third cause of CKD in the United States, whereas Australian data show that 73% of incident cases of CKD were attributable to diabetes, glomerulonephritis, hypertension, and polycystic kidney disease [28].

In order to evaluate the prevalence of glomerular diseases in this Brazilian cohort, all kidney biopsies with predominant glomerular involvement were sub grouped. A total of 810 biopsies were reclassified as glomerular diseases. Considering the prospective nature of the BKBR, the registry of newly diagnosed cases of glomerular diseases can be considered as incident cases. Thus, it seems reasonable to estimate that the incidence of glomerular diseases in Brazil corresponds to 0.38/100,000 inhabitants/year.

The most frequent glomerular diseases in the BKBR were lupus nephritis, IgAN and FSGS, resembling international reports [29–32]. It is recognized that IgAN is the most frequent glomerular disease in Asia and in some European countries [22, 29, 33] while FSGS has been the most common cause of primary glomerulopathy in the United States and Canada [11, 20, 31]. The emergence of IgAN ranking second in prevalence in the BKBR is different from previous reports [3, 5, 8]. This difference may rely on analyses related to the peculiar diversity of Brazilian regions. Alternatively, it should be mentioned that immunofluorescence techniques being routinely used in all kidney biopsies probably started in the last 15–20 years, which may explain the limited diagnosis of IgAN and other glomerulopathies so far. A possible explanation for ranking FSGS as the third most frequent glomerular disease in the BKBR is likely related to the classification of collapsing variant as a separate entity, currently included as AKD (APOL-1 Kidney Disease) [34].

Characteristics of glomerular diseases related to age, gender and clinical manifestations were similar to previous reports in other countries [16, 26, 35, 36].

Our study has some limitations. This was the first year of the registry's implementation, with data collected from the main nephrology centers in Brazil. However, the BKBR is an ongoing registry, currently more broadly assessed by Brazilian centers and nephrologists, with an increased number of biopsies being registered. As the first BKBR report, there is no follow-up data from patients yet. It should also be noted that the present data refers to 2021, the peak year of the COVID-19 pandemic in Brazil, which might have affected the number of biopsies performed that year.

In conclusion, the BKBR is a milestone in Brazilian national nephrological activity, including data from different regions of the country. Maintaining the registry, incorporating new centers not only in Brazil but expanding the registry for Latin America, and obtaining prospective data of registered patients represent the next challenges, aiming to provide accurate and precise data on the epidemiology of glomerular diseases in this part of the world.

## Supporting information

**S1 Table. Questionnaire available in the web-based registry system of the Brazilian Kidney Biopsy Registry (BKBR).**
(DOCX)

**S2 Table.** A. List of kidney diseases of this cohort grouped into six main categories. B. List of the diseases included in the category "Kidney diseases with Glomerular involvement".
(ZIP)

**S3 Table. Characteristics of kidney diseases in biopsies in elderly patients.**
(DOCX)

**S4 Table. Demographic, clinical and laboratory characteristics of patients with diabetes mellitus.**
(DOCX)

**S5 Table. List of collaborators.**
(DOCX)

## Acknowledgments

The authors are very grateful to Marcos Innocenti for the development of the registry web platform and for his excellent technical support. The authors are also grateful to Brazilian Society of Nephrology for the support for this registry.

## Author Contributions

**Conceptualization:** Irene L. Noronha, Rodrigo José Ramalho, Claudia Maria Costa de Oliveira, Marilia Bahiense-Oliveira, Fabricio Augusto Marques Barbosa, Jose de Resende Barros Neto, Osvaldo Merege Vieira-Neto, Precil Diego Miranda de Menezes Neves.

**Data curation:** Irene L. Noronha, Ronny Mitsuoka.

**Formal analysis:** Irene L. Noronha, Ronny Mitsuoka, Precil Diego Miranda de Menezes Neves.

**Methodology:** Irene L. Noronha, Rodrigo José Ramalho, Claudia Maria Costa de Oliveira, Marilia Bahiense-Oliveira, Fabricio Augusto Marques Barbosa, Jose de Resende Barros Neto, Precil Diego Miranda de Menezes Neves.

**Resources:** Osvaldo Merege Vieira-Neto.

**Writing – original draft:** Irene L. Noronha, Rodrigo José Ramalho, Claudia Maria Costa de Oliveira, Marilia Bahiense-Oliveira, Fabricio Augusto Marques Barbosa, Precil Diego Miranda de Menezes Neves.

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
