## [Decision Letter · Decision Letter 0]

6 May 2024

PONE-D-24-11117Implementation and first report of the Brazilian Kidney Biopsy RegistryPLOS ONE

Dear Dr. Noronha,

Thank you for submitting your manuscript to PLOS ONE. After careful consideration, we feel that it has merit but does not fully meet PLOS ONE’s publication criteria as it currently stands. Therefore, we invite you to submit a revised version of the manuscript that addresses the points raised during the review process.

We look forward to receiving your revised manuscript.

Kind regards,

Yavuz - Ayar

Academic Editor

PLOS ONE

Journal Requirements:

3. We note that your Data Availability Statement is currently as follows: All relevant data are reported in the paper and its Supporting information files.

5. One of the noted authors is a group or consortium "Brazilian Registry of Kidney Biopsies Working Group, Clinical 

Nephrology Department of the Brazilian Society of Nephrology". In addition to naming the author group, please list the individual authors and affiliations within this group in the acknowledgments section of your manuscript. Please also indicate clearly a lead author for this group along with a contact email address.

6. We note that Figure 1 in your submission contain map/satellite images which may be copyrighted. All PLOS content is published under the Creative Commons Attribution License (CC BY 4.0), which means that the manuscript, images, and Supporting Information files will be freely available online, and any third party is permitted to access, download, copy, distribute, and use these materials in any way, even commercially, with proper attribution. For these reasons, we cannot publish previously copyrighted maps or satellite images created using proprietary data, such as Google software (Google Maps, Street View, and Earth). For more information, see our copyright guidelines: http://journals.plos.org/plosone/s/licenses-and-copyright.

7. Thank you for stating the following in the Competing Interest section: ILN has received honoraria for Steering Committee roles, scientific presentations and/or advisory board attendance from Travere, Chinook, Vertex, Roche, and AstraZeneca. In addition, The George Institute for Global Health holds research contracts for trials in kidney disease. OMVN has received speaker’s honoraria from GSK, Takeda and AstraZeneca. All the other authors declare no conflict of interest.

We note that you received funding from a commercial source: Travere, Chinook, Vertex, Roche, AstraZeneca and GSK

Please include your amended Competing Interests Statement within your cover letter. We will change the online submission form on your behalf."

Additional Editor Comments:

Dear Author/s

Greetings

After the evaluations made by the reviewers, minor revision was decided.

Best regards

Reviewers' comments:

Reviewer's Responses to Questions

**Comments to the Author**

1. Is the manuscript technically sound, and do the data support the conclusions?

Reviewer #1: Yes

Reviewer #2: Yes

2. Has the statistical analysis been performed appropriately and rigorously? 

Reviewer #1: Yes

Reviewer #2: Yes

3. Have the authors made all data underlying the findings in their manuscript fully available?

Reviewer #1: Yes

Reviewer #2: Yes

4. Is the manuscript presented in an intelligible fashion and written in standard English?

Reviewer #1: Yes

Reviewer #2: Yes

5. Review Comments to the Author

Reviewer #1: I thank the authors for sharing their work. The manuscript is well written and explains the prevalence of renal pathologies from the Brazilian Kidney Biopsy Registry.

I have the following comments:

1) If this is the first article mentioning about the design and purpose of the BKBR registry, then the methods section would need to be elaborated.

2) Please mention more about how the data is collected. Are pathologists supposed to enter the data or will data be transferred from an electronic health record into the web interface.

3) Was the data (after entry by different centers) validated. Are there any measures to prevent incorrect data entry

4) The database is managed by MySQL system but could you also mention how the website was programmed (the language used etc).

5) Please provide details about how data can be requested by other researchers

6) Were there any laboratory data collected, such as creatinine and eGFR as there are also important.

Reviewer #2: I would like to congratulate the authors for the conduct of a nationwide registry in the 5th largest country in the world with such a wide geographic & demographic variations during COVID times.

1. Please confirm that the registry data collection is prospective while analysis is retrospective

2. The registry involves the patients with nephrotic syndrome & non-nephrotic range proteinuria. In practice we often see patients with nephrotic range proteinuria without nephrotic syndrome, such as in secondary FSGS. How such patients were analysed in the study?

3. In the results, discussion & Figure 4, the authors mention stage of CKD. I believe that they mean eGFR at the time of biopsy. Nephrotic syndrome due to MCD often responds to immunosuppression. It is not appropriate to call it as CKD. The authors can mention the eGFR value instead of stage of CKD

4. Fig 7 does not add any additional information than what is given by Fig 6b

5. Fig 9 can also be omitted as it does not an additional information than what is mentioned in the text. Gender difference in various glomerular disease is an established fact & the registry findings are congruent

6. Fig 11: As mentioned above, the mention of the stage of CKD is inappropriate. Also, the bar diagram used is not appropriate for the message that is being mentioned. I would suggest that the authors can give a range of eGFR at the time of biopsy with a median value in a tabular format or a similar figure could be constructed

7. Fig 12 is also not necessary- mentioned in the text

8. Fig 13 particularly interests me with the finding that nearly 44% of the diabetic patients showed primarily non-diabetic pathology. If possible, please mention the biopsy indications in the 145 diabetic patients, and what were the findings in those 64 patients. Also, can you give data of the biopsy finding as per the clinical indication

9. During discussion, estimation of glomerular disease incidence may need to be explained in a simpler way

6. PLOS authors have the option to publish the peer review history of their article (what does this mean?). If published, this will include your full peer review and any attached files.

Reviewer #1: No

Reviewer #2: **Yes: **Vipul Chakurkar

---

## [Author Response · Author response to Decision Letter 0]

21 Aug 2024

To 

Prof. Emily Chenette

Editor-in-chief of Plos One

Yavuz - Ayar

Academic Editor

Dear Prof. Chenette,

Thank you very much for your careful evaluation of our manuscript and the reviewers for their constructive comments. Enclosed, please find the new version of the manuscript with the changes highlighted in red font. The comments raised by the editor and reviewers have been incorporated into the revised version of the paper, and a detailed response to the reviewers is outlined below. As requested in item 7, an amended Competing Interests Statement was included (ILN and OMVN declare no relationship with employment, patents, products in development or marketed products with these Pharmaceutical Companies. This does not alter our adherence to PLOS ONE policies on sharing data and materials).

We believe we have appropriately addressed all points raised by the editors and the reviewers, and we hope our manuscript is suitable for publication in its present form.

We are looking forward to your decision.

Yours sincerely,

Prof. Irene L. Noronha, MD, PhD 

Full Professor of Nephrology 

Chief, Renal Division 

University of São Paulo 

REVIEWERS' COMMENTS TO THE AUTHOR

REVIEWER #1:

I thank the authors for sharing their work. The manuscript is well written and explains the prevalence of renal pathologies from the Brazilian Kidney Biopsy Registry.

Response: Thank you for your careful review of our manuscript. Your suggestions were fundamental to the improvement of our paper. We hope the responses below will clarify all the points you raised.

I have the following comments: 

1) If this is the first article mentioning about the design and purpose of the BKBR registry, then the methods section would need to be elaborated.

Response: According to your suggestion, we have better described the methods by adding all the raised suggestions. (Along page 4, highlighted in yellow)

2) Please mention more about how the data is collected. Are pathologists supposed to enter the data or will data be transferred from an electronic health record into the web interface.

Response: Data from patients of BKBR may be entered by pathologists or nephrologists from each site. We added a statement on methods as suggested (Page 4, 3rd paragraph).

3) Was the data (after entry by different centers) validated. Are there any measures to prevent incorrect data entry.

Response: This is an important point. In order to prevent incorrect data entry, the system was programmed to block some data, previously defined as possibly incorrect, such as creatinine values lower than 0,3 mg/dL or greater than 20 mg/dL, or the choose of both genders, or choosing more than one race, etc. The data added to the web-platform was regularly evaluated and validated by the statistician (RM), and incorrect or discrepant data was checked with each center that registered the patients for confirmation or correction. We added this information to the manuscript (Page 4, 3rd paragraph) 

4) The database is managed by MySQL system but could you also mention how the website was programmed (the language used etc).

Response: We described how the website was programmed, as requested (Page 4, 2nd paragraph). 

5) Please provide details about how data can be requested by other researchers

Response: We have described the data request protocol, as suggested (Page 4, 4th paragraph) 

6) Were there any laboratory data collected, such as creatinine and eGFR as there are also important.

Response: We collected the data of creatinine values. The eGFR is automatically calculated by the system, based on 2021 CKD-EPI, as described on Supplementary table 1.

REVIEWER #2:

I would like to congratulate the authors for the conduct of a nationwide registry in the 5th largest country in the world with such a wide geographic & demographic variations during COVID times.

Response: Thank you very much for your positive comments. We appreciated very much your comments. We hope the present answer will clarify all the points addressed.

1. Please confirm that the registry data collection is prospective while analysis is retrospective

Response: Thank you for your raised point. The Brazilian Kidney Biopsy Registry, indeed, is a prospective registry created in 2021 by Clinical Nephrology Department from Brazilian Society of Nephrology. In this manuscript we described the creation process of BKBR and the retrospective analysis of patients entered in 2021. 

2. The registry involves the patients with nephrotic syndrome & non-nephrotic range proteinuria. In practice we often see patients with nephrotic range proteinuria without nephrotic syndrome, such as in secondary FSGS. How such patients were analysed in the study?

Response: This is a relevant point. The presence of nephrotic-range proteinuria not associated with nephrotic syndrome indeed occurs frequently in the clinical setting. When the questionnaire was designed, the nephrologists members initially proposed a more complete questionnaire. However, in order to gain adherence to the registry, it was important to establish not only a friendly platform but also a relatively simple questionnaire, collecting the main relevant clinical and pathological information. For the next years, we have added will add not only this additional item as well as the value of proteinuria to be filled in, which will give us the opportunity to more accurately evaluate this data.

3. In the results, discussion & Figure 4, the authors mention stage of CKD. I believe that they mean eGFR at the time of biopsy. Nephrotic syndrome due to MCD often responds to immunosuppression. It is not appropriate to call it as CKD. The authors can mention the eGFR value instead of stage of CKD.

Response: Thank you for your observation. In 2002, the Kidney Disease Outcomes Quality Initiative (KDOQI) presented a definition and classification of CKD, which was accepted by the international community in 2005. In 2012, the KDIGO CKD guidelines, based on the KDOQI definition and classification, reinforced the definition of CKD incorporating a persistent reduction in eGFR and markers of kidney damage besides modifying the staging and classification system. The definition, staging, and classification of CKD proposed by the KDIGO 2012 CKD guidelines have been widely accepted and implemented worldwide. The update KDIGO Guidelines 2024, maintained the definition and the classification. The point raised by the reviewer makes sense, similarly to other discussion such as eGFR > 90 ml/min already being defined as a stage 1 of CKD. However, considering that KDIGO classification of CKD is worldwide accepted and that Figure 4 presents both information (the eGFR and the corresponding CKD stages), we considered to maintain figure 4.

4. Fig 7 does not add any additional information than what is given by Fig 6b.

Response: Thank you for signalling this question, that indicate we did not present these results properly, raising misunderstanding. 

We reformulated the titles of figure 6 and Fig 7, highlighting the different results presented in each figure. Figure 6 presents the results of histological diagnosis of all kidney biopsies of this cohort classified into six main categories, whereas Figure 7 shows the frequencies of diseases with glomerular involvement in kidney biopsies in this cohort

Figure 6. Histological diagnosis of all kidney biopsies of this cohort classified into six main categories. 6A) Frequencies of kidney biopsies diagnosis grouped into the six main categories. 6B) Detailed numbers and percentages of all specific diagnoses, separated according the categories 

5. Fig 9 can also be omitted as it does not an additional information than what is mentioned in the text. Gender difference in various glomerular disease is an established fact & the registry findings are congruente

Response: We agree with the reviewer. Figure 9 was withdrawn. 

6. Fig 11: As mentioned above, the mention of the stage of CKD is inappropriate. Also, the bar diagram used is not appropriate for the message that is being mentioned. I would suggest that the authors can give a range of eGFR at the time of biopsy with a median value in a tabular format or a similar figure could be constructed

Response: Thank you for your observation. We have answered this concept and CKD definition in question 3. However, following the reviewer´s suggestion, we changed the graphic, including the eGFR ranges. 

7. Fig 12 is also not necessary- mentioned in the text

Response: The reviewer is correct We usually avoid presenting the data in the text and simultaneously presenting it as figure. In this case, we thought it would be interesting to show as a figure to point out the different percentages of lupus nephritis classes diagnosed in the registry. However, following the reviewer’s suggestion, Figure 12 was withdrawn.

8. Fig 13 particularly interests me with the finding that nearly 44% of the diabetic patients showed primarily non-diabetic pathology. If possible, please mention the biopsy indications in the 145 diabetic patients, and what were the findings in those 64 patients. Also, can you give data of the biopsy finding as per the clinical indication

Response: Clinical and laboratory characteristics responsible for biopsy indications in the 145 patients with diabetes mellitus were included as Supplemental table 3. The biopsy findings of the 64 diabetic patients without evidence of diabetic nephropathy on kidney biopsy were also presented in the manuscript (page 12)

9. During discussion, estimation of glomerular disease incidence may need to be explained in a simpler way

Response: The sentences related to the incidence of glomerular diseases were reformulated (page 14).

---

## [Decision Letter · Decision Letter 1]

7 Oct 2024

Implementation and first report of the Brazilian Kidney Biopsy Registry

PONE-D-24-11117R1

Dear Dr. Noronha,

We’re pleased to inform you that your manuscript has been judged scientifically suitable for publication and will be formally accepted for publication once it meets all outstanding technical requirements.

Kind regards,

Junzheng Yang

Academic Editor

PLOS ONE

Additional Editor Comments (optional):

Reviewers' comments:

Reviewer's Responses to Questions

**Comments to the Author**

1. If the authors have adequately addressed your comments raised in a previous round of review and you feel that this manuscript is now acceptable for publication, you may indicate that here to bypass the “Comments to the Author” section, enter your conflict of interest statement in the “Confidential to Editor” section, and submit your "Accept" recommendation.

Reviewer #1: All comments have been addressed

Reviewer #2: All comments have been addressed

2. Is the manuscript technically sound, and do the data support the conclusions?

Reviewer #1: Yes

Reviewer #2: Yes

3. Has the statistical analysis been performed appropriately and rigorously? 

Reviewer #1: Yes

Reviewer #2: I Don't Know

4. Have the authors made all data underlying the findings in their manuscript fully available?

Reviewer #1: Yes

Reviewer #2: Yes

5. Is the manuscript presented in an intelligible fashion and written in standard English?

Reviewer #1: Yes

Reviewer #2: Yes

6. Review Comments to the Author

Reviewer #1: The authors have made all the necessary changes to the manuscript and also addressed other potential concerns. I do not have anything further to add.

Reviewer #2: Thank you for your response. I am happy to know the authors are open to comments and ready to incorporate the suggestions in the registry

7. PLOS authors have the option to publish the peer review history of their article (what does this mean?). If published, this will include your full peer review and any attached files.

Reviewer #1: No

Reviewer #2: **Yes: **Dr Vipul Chakurkar

---

## [Editor Report · Acceptance letter]

23 Oct 2024

PONE-D-24-11117R1 

PLOS ONE

Dear Dr. Noronha, 

I'm pleased to inform you that your manuscript has been deemed suitable for publication in PLOS ONE. Congratulations! Your manuscript is now being handed over to our production team.

Kind regards, 

on behalf of

Director Junzheng Yang 

Academic Editor

PLOS ONE